# Towards Reasoning-Aware Explainable VQA

**Rakesh Vaideeswaran**[*,1], **Feng Gao**[2], **Abhinav Mathur**[2], **Govind Thattai**[2]
[1] University of Illinois, Urbana-Champaign
[2] Amazon Alexa AI
rmahesh3@illinois.edu, {fenggo,mathuabh,thattg}@amazon.com

## Abstract

The domain of joint vision-language understanding, especially in the context of reasoning in Visual Question Answering (VQA) models, has garnered significant attention in the recent past. While most of the existing VQA models focus on improving the accuracy of VQA, the way models arrive at an answer is oftentimes a black box. As a step towards making the VQA task more explainable and interpretable, our method is built upon the SOTA VQA framework [1] by augmenting it with an end-to-end explanation generation module. In this paper, we investigate two network architectures, including Long Short-Term Memory (LSTM) and Transformer decoder, as the explanation generator. Our method generates human-readable textual explanations while maintaining SOTA VQA accuracy on the GQA-REX (77.49%) and VQA-E (71.48%) datasets. Approximately 65.16% of the generated explanations are approved by humans as valid. Roughly 60.5% of the generated explanations are valid and lead to the correct answers.

## 1  Introduction

Problems involving joint vision-language understanding are gaining more attention in both Computer Vision (CV) and Natural Language Processing (NLP) communities. In recent years, complex reasoning problems in the vision-language domain have been in the spotlight. In the classic Visual Question Answering (VQA) problem, reasoning has been highly involved. In [2, 3, 4], a model needs to reason over spatial and quantificational relationships within an image-question pair. [5] incorporates spatial-temporal reasoning as well as domain-specific knowledge. A more challenging setting, such as [6, 7], requires the capability to make use of external knowledge to perform reasoning in the vision-language domain.

We see impressive improvements in VQA accuracy in [8, 9, 10, 11] in both the stock setting and its variants, thanks to the large-scale pre-trained models in both single modality and multi-modality. However, we barely pay attention to how a model reaches an answer given an image-question pair. Let's take a look at Figure 1 as an illustrative example. The ground-truth answer to the question is straightforward, and information from the image is sufficient to answer the question. There are three different types of VQA models: **Model type 1** predicts the correct answer without providing any evidence as to how it was achieved. **Model type 2** answers the question correctly and provides a caption that summarizes the image. Unfortunately, the caption fails to unveil

**Q**: Is the stovelight on?   **GT Ans:** Yes
**M1**, **M2**, **M3**: Yes
**C**: There is a stove and a bunch of knives
**E**: The space under the hood is brighter than the surrounding area.

Figure 1: A VQA example shows the importance of an explanation that leads to the correct answer.

---

⋆ This author completed this work during his internship at Amazon.

2022 Trustworthy and Socially Responsible Machine Learning (TSRML 2022) co-located with NeurIPS 2022.

the reasoning chain behind predicting the correct answer to the question. **Model type 3** successfully generates a logically self-contained explanation which corresponds to the correct answer. Both **Model type 1** and **Model type 2** cover most of the SOTA VQA models. Surprisingly, very few models exist that are similar to **Model type 3**. Motivated by examples like in Figure 1, we investigate the following two open topics in this paper: (i) whether a VQA model can generate a human-readable explanation while maintaining VQA accuracy; (ii) how good are the generated explanations and how should they be evaluated? Our contribution is two-fold:

- We present easy-to-implement methods on top of a SOTA VQA framework which maintains VQA accuracy while generating human-readable textual explanations.

- We show both quantitative experimental results and human-studies of the proposed explainable VQA method. Our experiments illustrate the urgency of proposing new metrics to evaluate the predicted explanations in vision-language reasoning problems such as VQA.

## 2    Related Work

**Reasoning in VQA**    As an end-to-end task, VQA [4] and its variants have been well explored, and various models have kept achieving better performance along the way. Built on top of the original setting, complex reasoning tasks are heavily involved. [4, 12] introduce quantificational reasoning into the setting. [3, 13, 2] highlight the importance of fine spatial and compositional reasoning in the VQA problem. While the above datasets limit the reasoning domain within the image, [6, 14] propose a visual language task that requires external knowledge, sometimes even domain-specific knowledge, to answer the question. In [15], the emphasis is on the logical entailment problem. Recent methods such as [8, 9, 10] take advantage of the unprecedented amount of vision-language data and large size of models, achieving SOTA performances on the above VQA datasets. As the reasoning problem in VQA is becoming more and more complicated, it is urgent to have an interpretable way to analyze and diagnose the model and measure its reliability.

**Explainable VQA and Metrics**    Very few SOTA VQA works investigate model explainability, especially in the age of big data and big models. [16] is one of the standard datasets that focuses on explainability. A similarity score between the question and the image caption is computed to check for question-relevant captions. The caption is then used to generate an explanation that is relevant to the question-answer pair. [17, 18] use image attributes and captions to provide a naive version of the explanation to the answer. Some works [19, 11] make use of textual knowledge from external sources to improve the interpretability of the model. However, such external knowledge is not always able to provide direct evidence to the answer. The neural-symbolic framework [20] is also applied in the VQA domain since it is naturally more interpretable than the pure deep learning based methods. More works [21, 22] have recently been proposed for enhancing the explainability of the VQA problem using either natural or synthetic data. Another topic that is not well-studied in explainable VQA is the evaluation of explanations. In [16, 22], conventional NLP metrics such as ROUGE, BLEU scores are used to measure the quality of the generated explanation. In contrast to [16], [21] doesn't use a caption as the explanation, instead it uses tokens representing a bounding box in the image to replace key parts in the scene graph.

## 3    Methodology

In this section, we describe our method in detail. Please refer to Figure 2 for an overview of the model flow and architecture. The proposed method consists of two major components: (i) coarse-to-fine visual language reasoning for VQA and (ii) explanation generation module.

### 3.1    Extracting Features and Predicates

A pre-trained Faster-RCNN model[23] is used to extract features for each Region of Interest (RoI) in the image $I$. The image features are denoted as $f_I$. Similarly, a Faster-RCNN model is also used to extract objects and attributes that form the image predicates. We generate the Glove embedding[24] for each word in the set of image predicates, denoted as $p_I$. The words in the question $Q$ are also encoded using Glove embeddings. The question embeddings are then passed through a GRU to

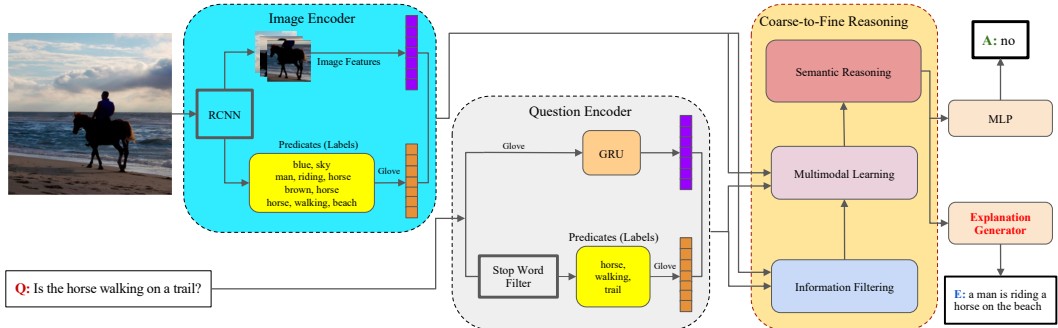

Figure 2: An overview of coarse-to-fine VQA with explanation generation.

extract sequential features $f_Q$. Together with this, question predicates are extracted by passing the question through a stop word filter. The stop words not only consist of words from NLTK[25], but also include those words in questions that occur less frequently (threshold=10). Each question predicate is then encoded with Glove embedding. Question predicates are denoted as $p_Q$.

## 3.2 Coarse-to-Fine Reasoning for VQA

VQA can be generally formulated as $(I, Q) \rightarrow \mathbf{a}$, where $a \in \mathcal{A}$ and $\mathcal{A}$ is the set of answers. Usually, the answer set $\mathcal{A}$ is filtered by a frequency threshold from the annotated answers. The coarse-to-fine reasoning framework can be formalized as:

$$
\begin{aligned}
\mathbf{a}^* &= arg \max_a \mathbf{CFR}_\theta(\mathbf{a}|f_I, p_I, f_Q, p_Q) \\
&= arg \max_a \mathbf{SR}(\mathbf{a}|\mathbf{IF}(f_I, p_I, f_Q, p_Q), \mathbf{MM}(f_I, p_I, f_Q, p_Q))
\end{aligned}
\tag{1}
$$

where $\mathbf{CFR}_\theta$ is an end-to-end module with learnable parameter $\theta$. It consists of three different modules, including an information filtering module $\mathbf{IF}$, a multimodal learning module $\mathbf{MM}$, followed by a semantic reasoning module $\mathbf{SR}$.

**Information Filtering**  The extracted features may be noisy and contain incorrect information as they are extracted from pre-trained models. This module helps remove unnecessary information and aids in understanding the importance of RoIs in images for each question.

**Multimodal Learning**  Bilinear Attention Networks are used to learn features at both coarse-grained and fine-grained levels. The coarse-grained module works with image and question features and predicates and produces a joint representation at the coarse-grained level. The fine-grained module learns the correlation between the filtered image and the question information and learns a joint representation at the fine-grained level.

**Semantic Reasoning**  This module learns selective information from both the coarse-grained and fine-grained module outputs. The joint embedding from this module is then fed into a multi-layer perceptron to perform answer prediction and to the explanation module for explanation generation.

## 3.3 Explanation Generation

The joint embedding from the semantic reasoning module is used to train an explanation generator with ground-truth explanations as supervision. The VQA backbone is augmented with the explanation generation module. Two architectures are evaluated for explanation generation: (i) Long Short-Term Memory (LSTM), (ii) Transformer Decoder. The LSTM architecture used consists of 2 layers, with an input dimension of 768. The Transformer Decoder architecture has an input dimension of 768, and consists of 8 attention heads. In both cases, the input is a joint embedding and is trained using ground-truth explanations from the dataset (discussed in the following section) using cross-entropy loss for each word. Suppose we have an explanation $\mathbf{E} = (w_1, ..., w_i, .., w_l)$, where $w_i \in \mathbb{V}$, the vocabulary and $l$ is the length of the explanation. The explanation can therefore be represented as a sequence of one-hot encoded vectors. The loss function is therefore given by:

$$L_{expl} = -\frac{1}{l \cdot |\mathbb{V}|} \cdot \sum_{i=1}^{l} \sum_{k=1}^{|\mathbb{V}|} y_{i,k} \cdot \log(p(w_{i,k})) \qquad (2)$$

where $y_{i,k}$ is the one-hot vector for the $i^{th}$ word in the ground-truth explanation and $p(w_{i,k})$ is the probability of the $k^{th}$ word in $\mathbb{V}$ at the $i^{th}$ time step. We also make use of teacher enforcing to train the explanation module with autoregressive cross-entropy loss.

## 4 Experiments

### 4.1 Datasets and Evaluation

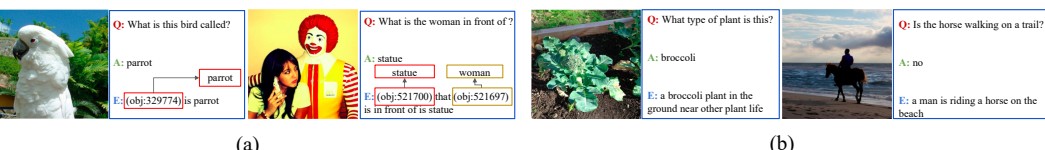

Figure 3: Examples from (a) GQA-REX dataset, (b) VQA-E dataset.

As discussed in section 2, there are a limited number of datasets that come with annotated explanations along with answers. Owing to the large dataset size, we perform our experiments on the **GQA-REX** and **VQA-E** datasets, although they have their own set of limitations. In this section, we measure the accuracy of the predicted answer and the quality of the generated explanation. To evaluate the predicted answer, we use the **VQA score** as the metric. Unfortunately, we don't have accurate metrics for explanation evaluation. Therefore, we report qualitative results from a **human-study** as well as quantitative results using conventional NLP metrics such as **ROUGE** and **BLEU**.

**GQA-REX**   contains explanations for almost 98% of the samples in the GQA-balanced dataset. It contains around 1.04M question-answer (QA) pairs spanning across 82K images, with annotated explanations (1 explanation per QA pair). However, the explanations are consistent with the reasoning framework proposed in [21] and are therefore not completely in human-readable form (Refer Figure 3(a)). Although the explanations can be converted to human readable form using information from scene graphs, there exist instances of grammatical inaccuracy.

**VQA-E**   contains explanations for around 40% of the QA pairs in the VQA2.0 dataset (1 explanation per QA pair). The explanations are generated by comparing the similarity scores between the caption candidates and the ground-truth question-answer pair. It is, therefore, not surprising that the explanations seem more like captions of the image that contains the answer. Figure 3(b) illustrates a couple of examples from the VQA-E dataset.

### 4.2 VQA Experimental Results

| Dataset | Expl. Model | $\alpha$ | VQA score |
|---------|-------------|----------|-----------|
| VQA-E[16] | N/A | N/A | 71.48 |
| | LSTM | 0.25 | 71.36 |
| | LSTM | 0.50 | **71.55** |
| | LSTM | 0.75 | 71.53 |
| | LSTM | 1.0 | 71.32 |
| | Transformer | 0.50 | 71.46 |
| GQA-REX[21] | N/A | N/A | 77.49 |
| | LSTM | 0.25 | 75.08 |
| | LSTM | 0.50 | 77.16 |
| | LSTM | 1.0 | **77.33** |
| | Transformer | 0.50 | 77.06 |

Table 1: VQA scores of the predicted answers from our method on VQA-E and GQA-REX validation datasets.

We use the CFRF[1] model as the backbone and augment it with an explanation generation module based on (i) LSTM and (ii) Transformer Decoder. The baseline model is trained without any explanations as supervision. Since our goal is to generate explanations while maintaining the VQA performance, we incorporate both the loss for VQA answer and the supervision from the ground-truth explanation. In order to investigate the impact of two different training signals, we design the loss function for the end-to-end training as follows: $L = \alpha L_{ans} + (1 - \alpha) L_{expl}$, where

$L_{ans}$ is the cross-entropy loss between the predicted answer and the ground truth answer, $L_{expl}$ is the loss function of the explanation generation module, as represented by Equation 2, and $\alpha \in [0, 1]$ is the balance factor. As shown in Table 1, our methods successfully maintain the VQA scores while generating textual explanations.

## 4.3 Results of Generated Explanations

**Quantitative Results** We use the explanations generated by the CFRF+LSTM model, corresponding to $\alpha = 0.75$ (Refer Table 1). The results of BLEU-1 and ROUGE scores are presented. Note that ROUGE scores are F1 scores. As shown in Table 2, although our method outperforms the baseline, the absolute scores are only satisfactory.

| Dataset | Model | BLEU-1 | ROUGE-1 | ROUGE-2 | ROUGE-L |
|---------|-------|--------|---------|---------|---------|
| VQA-E val | Baseline[16] | 0.268 | - | - | 0.249 |
| | CFRF+LSTM | 0.33 | 0.364 | 0.117 | 0.325 |

Table 2: Quantitative evaluation of the generated explanations on VQA-E validation set.

As mentioned in section 2, in the VQA domain, there is no standard common practice to quantitatively evaluate generated explanations. Although both VQA-E and GQA-REX suggest using conventional NLP metrics such as ROUGE and BLEU scores to evaluate generated explanations, it is not ideal. These metrics are particularly designed for string matching in the form of overlapping n-grams. Figure 4 illustrates why such metrics are practically unreliable.

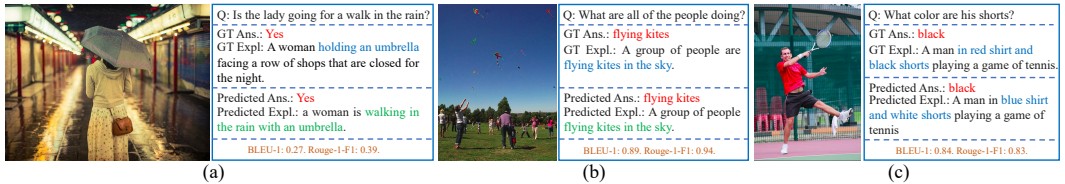

Figure 4: Problem with using string matching metrics to evaluate generated explanations in VQA.

In Figure 4(a), our model predicts the correct answer. However, according to the string matching metrics, the quality of the explanation is poor. In fact, interestingly, both the generated explanation and the ground-truth explanation in Figure 4(a) are annotated as valid by human annotators. On the other hand, the generated explanation in Figure 4(b) is almost identical to the ground truth, and both of them are approved by human subjects as valid explanations for the answer. In Figure 4(c), even though the predicted explanation is wrong, the string matching score is very high. These examples lead to the following conclusion: we need to find a more reliable metric to evaluate generated explanations for the VQA problem.

**Human Study Setup** Since no mature quantitative metrics are available, we introduce humans into the loop. We conducted a human subject study using Amazon Mechanical Turk (AMT). The goal of our subject study is to evaluate the quality of the explanation from human annotation. One example of the human intelligence task (HIT) is shown in Figure 5:

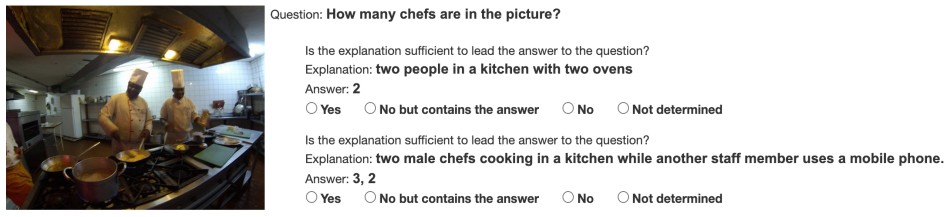

Figure 5: An example of a HIT in the human study.

Given an image-question pair from the VQA-E validation set, we designed two questions for the annotators. Both questions are the same, asking whether an explanation leads to the answer. But the contexts are different. In the first question, both the explanation and the answer are generated by our

| Context | Yes | No, but contains the Ans. | No | Not determined |
|---|---|---|---|---|
| predicted [total] | 56.46% | 9.02% | 34.12% | 0.4% |
| predicted [unique] | 65.16 % | 2.01% | 32.8% | 0.03% |
| ground-truth [total] | 83.90% | 5.12% | 10.98% | 0% |
| ground-truth [unique] | 93.12% | 0.57% | 6.31% | 0% |

Table 3: Statistics of the raw human annotation data. It contains 4735 unique examples from the VQA-E validation set. Each job is distributed to 3 different annotators to eliminate potential bias.

model. In the second question, we provide the ground-truth explanation and answer. The subjects have the same set of four options to choose from in both cases. They are as follows: (i) Yes; (ii) No, but contains the answer; (iii) No; (iv) Not determined. Annotators have no idea which context is the ground truth. Specifically, option (ii) "No, but contains the answer" means the explanation contains a sub-string that matches the predicted answer but it does not lead to the answer. Option (iv) "Not determined" means the explanation leads to the answer, but the reasoning chain may be contradictory.

**Human Approved Results** We randomly selected 4735 unique image-question pairs from the VQA-E validation set for the human study. Each image-question pair makes up one HIT with the same setting as in Figure 5. In order to eliminate individual bias, we assigned each HIT to three different workers. Therefore, in total, we received $4735 \times 3 = 14205$ responses from 111 subjects. The raw distribution of subject annotations for all the 14205 responses (predicted [total] and ground-truth [total]) is shown in Table 3. From the total set of responses, questions for which there is no consensus among the three annotators (all three responses are different) are discarded (869 out of 4735). Following this, we calculate the vote using mode, i.e., a majority vote for the unique HITs. Among the 3866 unique HITs, **65.16%** of the generated explanations lead to the predicted answers, while 2.01% of them contain the answers but make no sense. 32.8% of the generated explanations fail to make connections with the predicted answers. On the other hand, 93.12% of the ground-truth explanations lead to ground-truth answers. According to [16], because the ground-truth explanations are selected by comparing the similarity between the question-ground-truth-answer pair and the caption candidates, most of them are valid.

| | Predicted | | Ground-truth | |
|---|---|---|---|---|
| | Valid Expl. | Invalid Expl. | Valid Expl. | Invalid Expl. |
| Correct Ans. | **56.39%** | 23.77% | 93.11% | 6.88% |
| Wrong Ans. | 8.77% | 11.05% | - | - |

Table 4: Ratio of valid/invalid explanation based on the correctness of the predicted answer.

Besides raw annotations, we also provide a more straightforward result, as shown in Table 4. Among the 3866 unique HITs, we find that in 56.39% of the cases, our model can predict both the correct answer as well as generate valid explanations. 23.77% of the explanations are not valid, although the predicted answers are correct. It may either make no sense or contain the answer in it, albeit with little significance. Only in 8.77% of the cases, our model generates a good explanation but leads to a wrong answer. On the other hand, we also observe that 6.88% of the ground-truth explanations are not reasonable. Therefore, our model is able to answer questions correctly and also generate valid explanations approximately **60.5%** of the time.

## 5 Conclusion and Future Work

We explore the task of Explainable Visual Question Answering (Explainable-VQA). We leverage the Coarse-to-Fine reasoning framework as the VQA backbone and augment it with an explanation generation module using two architectures: LSTM and Transformer Decoder. Our model generates an explanation along with an answer while also maintaining close to SOTA VQA performance. We conduct both objective experiments and a human study to evaluate the generated explanation, pointing out the urgency of proposing new metrics for explainable VQA.

**Future Work** We plan to improve the quality of generated explanations as well as leverage them to increase VQA accuracy. We urge proper metrics to evaluate explanations for the VQA problem.

**Acknowledgement** We thank Nguyen et al., the authors of [1] for providing us with the features and predicates for the VQA 2.0 dataset and helping with answering all queries in a timely manner.

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

# Appendix A  Examples for Predicted Answers and Explanations

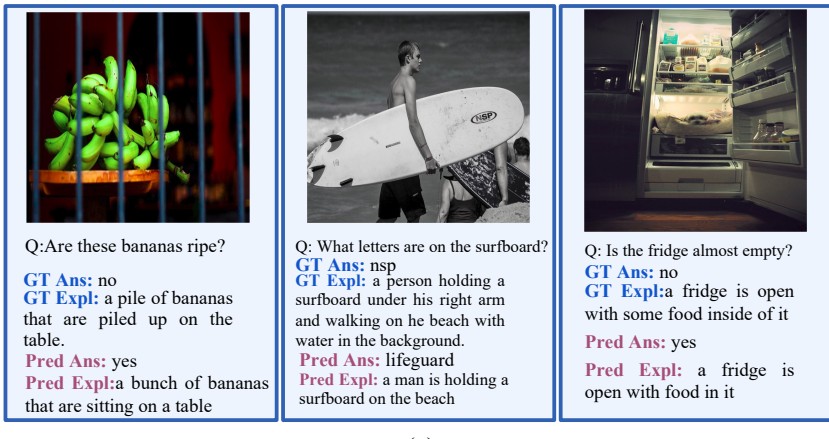

(a)

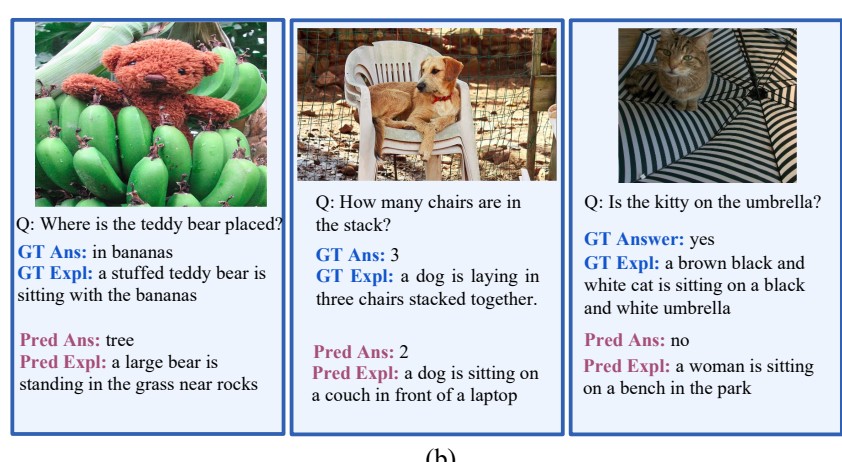

(b)

Figure 6: (a) 3 examples for incorrectly predicted answer but correct explanation. (b) 3 examples for incorrectly predicted answer and incorrect explanation.

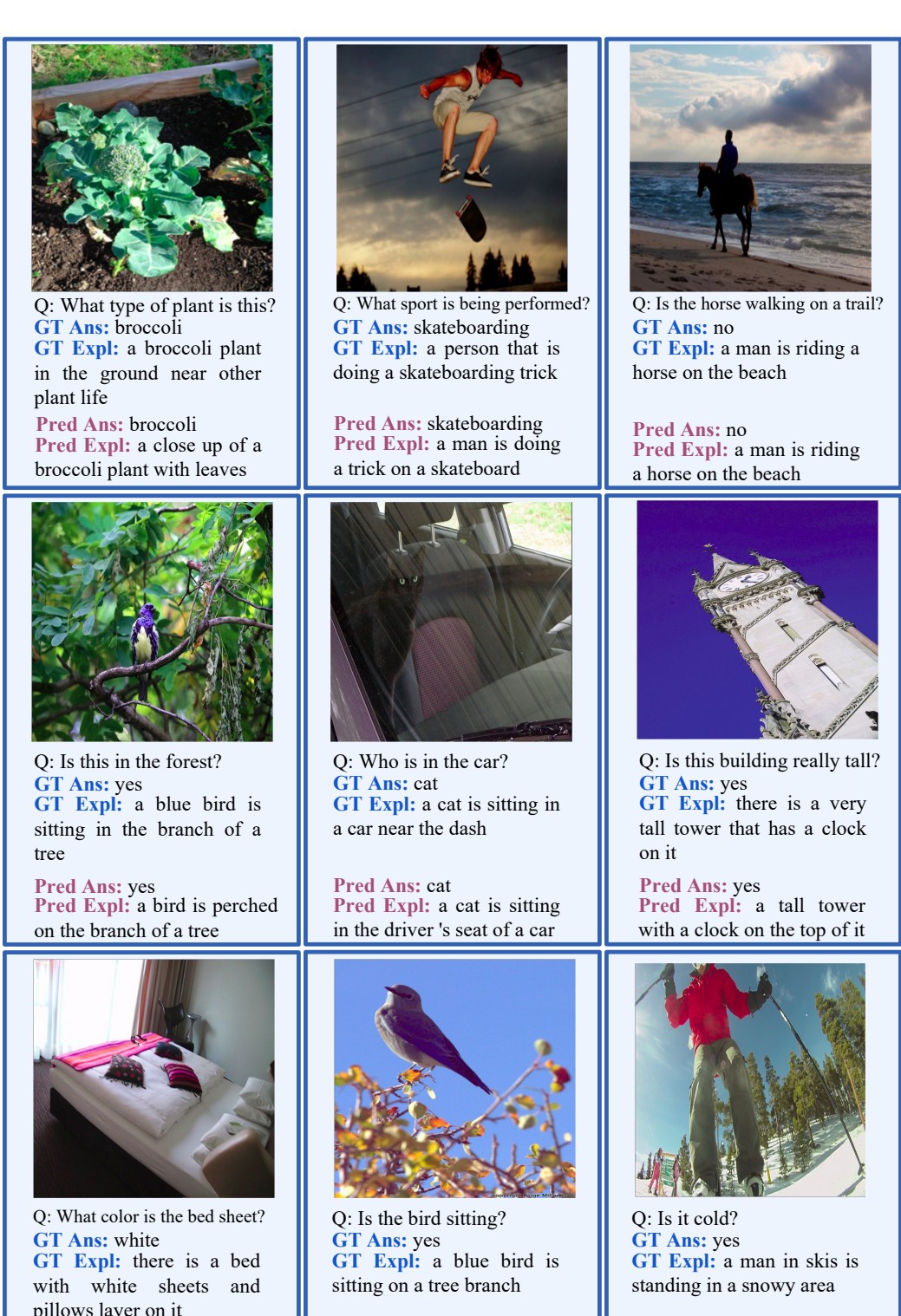

Q: What type of plant is this?
**GT Ans:** broccoli
**GT Expl:** a broccoli plant in the ground near other plant life

**Pred Ans:** broccoli
**Pred Expl:** a close up of a broccoli plant with leaves

Q: What sport is being performed?
**GT Ans:** skateboarding
**GT Expl:** a person that is doing a skateboarding trick

**Pred Ans:** skateboarding
**Pred Expl:** a man is doing a trick on a skateboard

Q: Is the horse walking on a trail?
**GT Ans:** no
**GT Expl:** a man is riding a horse on the beach

**Pred Ans:** no
**Pred Expl:** a man is riding a horse on the beach

Q: Is this in the forest?
**GT Ans:** yes
**GT Expl:** a blue bird is sitting in the branch of a tree

**Pred Ans:** yes
**Pred Expl:** a bird is perched on the branch of a tree

Q: Who is in the car?
**GT Ans:** cat
**GT Expl:** a cat is sitting in a car near the dash

**Pred Ans:** cat
**Pred Expl:** a cat is sitting in the driver 's seat of a car

Q: Is this building really tall?
**GT Ans:** yes
**GT Expl:** there is a very tall tower that has a clock on it

**Pred Ans:** yes
**Pred Expl:** a tall tower with a clock on the top of it

Q: What color is the bed sheet?
**GT Ans:** white
**GT Expl:** there is a bed with white sheets and pillows layer on it

**Pred Ans:** white
**Pred Expl:** a bed with a white comforter and pillows on it

Q: Is the bird sitting?
**GT Ans:** yes
**GT Expl:** a blue bird is sitting on a tree branch

**Pred Ans:** yes
**Pred Expl:** a bird is sitting on the branch of a tree

Q: Is it cold?
**GT Ans:** yes
**GT Expl:** a man in skis is standing in a snowy area

**Pred Ans:** yes
**Pred Expl:** a man is standing in the snow with skis and ski poles

Figure 7: 9 examples for correctly predicted answer and correct explanation.

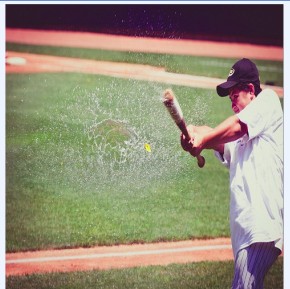

Q: What game is he playing?
**GT Ans:** baseball
**GT Expl:** a baseball player is swinging a bat and some grass

**Pred Ans:** baseball
**Pred Expl:** a baseball player is swinging his bat

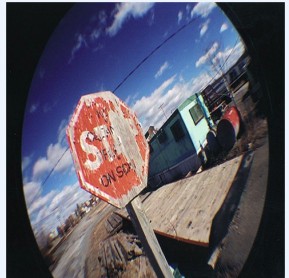

Q: Is this stop sign red?
**GT Ans:** yes
**GT Expl:** a red stop sign sitting on top of a wooden post.

**Pred Ans:** yes
**Pred Expl:** a red stop sign sitting in the middle of a road

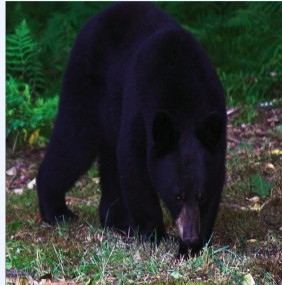

Q:What color is the animal?
**GT Ans:** black
**GT Expl:** a black bear is standing outdoors in the wild.

**Pred Ans:** black
**Pred Expl:** a black bear is walking around in the woods

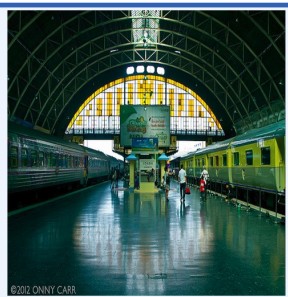

Q: How many trains are there?
**GT Ans:** 2
**GT Expl:** two trains are on tracks in a commuter strain station with people standing on the platform between them.
**Pred Ans:** 2
**Pred Expl:** two trains are parked on the tracks at a station

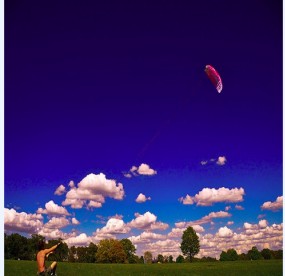

Q: Where is the kite?
**GT Ans:** sky
**GT Expl:** a man sitting in a field flying a kite in the blue sky.

**Pred Ans:** sky
**Pred Expl:** a man is flying a kite in the sky high

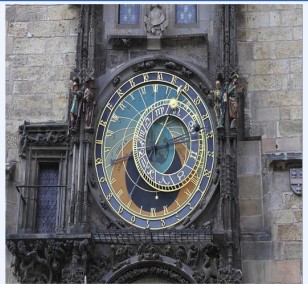

Q: What type of numbers are used?
**GT Ans:** roman numerals
**GT Expl:** a clock tower with roman numerals and a sun dial.

**Pred Ans:** roman numerals
**Pred Expl:** a clock on a building with roman numerals and a clock

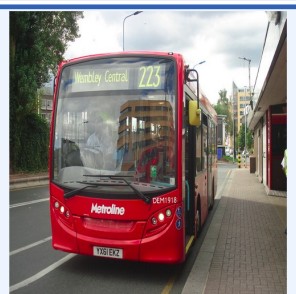

Q: What color is the bus?
**GT Ans:** red
**GT Expl:** a red bus parked in front of a building near a street.

**Pred Ans:** red
**Pred Expl:** a red and white bus parked in front of a building

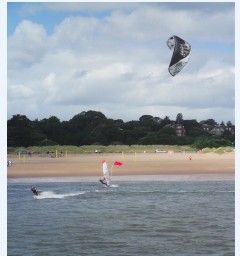

Q: Is the kite lifting in the wind?
**GT Ans:** yes
**GT Expl:** folks flying a kite in the water of a nice beach.
**Pred Ans:** yes
**Pred Expl:** a kite flying in the sky over a body of water

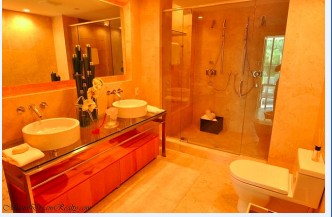

Q: How many sinks are there?
**GT Ans:** 2
**GT Expl:** modern bathroom with two sinks a toilet and a shower.

**Pred Ans:** 2
**Pred Expl:** a bathroom with two sinks and a mirror above it

Figure 8: 9 examples for correctly predicted answer and correct explanation.

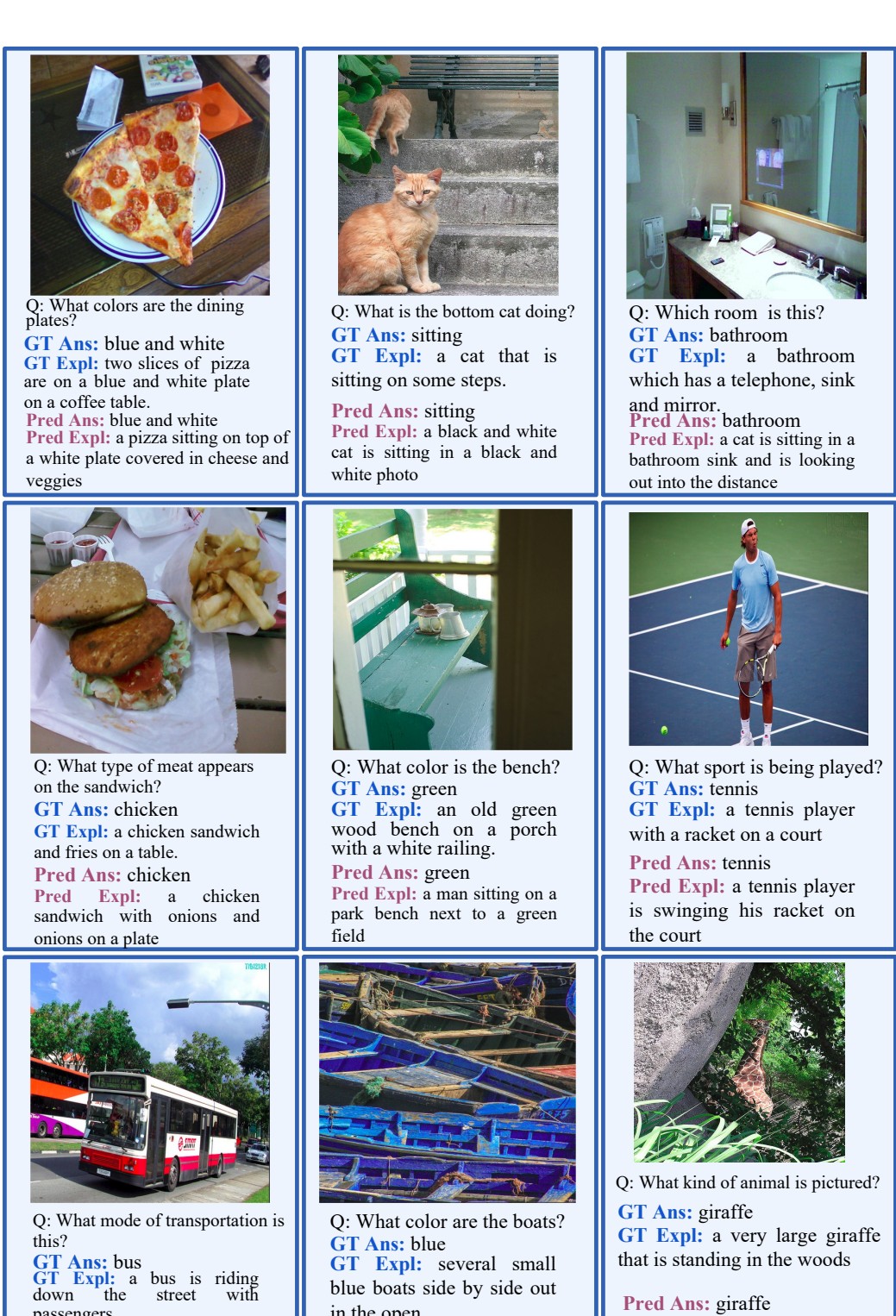

Q: What colors are the dining plates?
**GT Ans:** blue and white
**GT Expl:** two slices of pizza are on a blue and white plate on a coffee table.
**Pred Ans:** blue and white
**Pred Expl:** a pizza sitting on top of a white plate covered in cheese and veggies

Q: What is the bottom cat doing?
**GT Ans:** sitting
**GT Expl:** a cat that is sitting on some steps.
**Pred Ans:** sitting
**Pred Expl:** a black and white cat is sitting in a black and white photo

Q: Which room is this?
**GT Ans:** bathroom
**GT Expl:** a bathroom which has a telephone, sink and mirror.
**Pred Ans:** bathroom
**Pred Expl:** a cat is sitting in a bathroom sink and is looking out into the distance

Q: What type of meat appears on the sandwich?
**GT Ans:** chicken
**GT Expl:** a chicken sandwich and fries on a table.
**Pred Ans:** chicken
**Pred Expl:** a chicken sandwich with onions and onions on a plate

Q: What color is the bench?
**GT Ans:** green
**GT Expl:** an old green wood bench on a porch with a white railing.
**Pred Ans:** green
**Pred Expl:** a man sitting on a park bench next to a green field

Q: What sport is being played?
**GT Ans:** tennis
**GT Expl:** a tennis player with a racket on a court
**Pred Ans:** tennis
**Pred Expl:** a tennis player is swinging his racket on the court

Q: What mode of transportation is this?
**GT Ans:** bus
**GT Expl:** a bus is riding down the street with passengers.
**Pred Ans:** bus
**Pred Expl:** a bus is parked in the parking lot while a passenger boards

Q: What color are the boats?
**GT Ans:** blue
**GT Expl:** several small blue boats side by side out in the open.
**Pred Ans:** blue
**Pred Expl:** a couple of small boats are on the blue water

Q: What kind of animal is pictured?
**GT Ans:** giraffe
**GT Expl:** a very large giraffe that is standing in the woods
**Pred Ans:** giraffe
**Pred Expl:** a giraffe is standing in the grass near a fence

Figure 9: 9 examples for correctly predicted answer but incorrect explanation.

