# OpenReview forum: "Towards Reasoning-Aware Explainable VQA"
_NeurIPS.cc/2022/Workshop/TSRML — TSRML2022_

### Official Review · Reviewer_vsYb · 2022-10-17

**Overall Rating:** 4

**Summary:**

This paper studies the problem of explainable Visual Question Answering (VQA). In the VQA setting, an ML model is given an image and a question about the image as input, and produces an answer as output. In order to develop an explainable VQA model, the authors augment the state-of-the-art VQA model with an end-to-end "explanation generation module". The authors investigate two architectures for the explanation generation module: one based on LSTM and another based on Transformer Decoder. The models are empirically evaluated on the GQA-REX and VQA-E datasets. In order to evaluate the quality of the explanation generated, the authors conducted a human study on Amazon Mechanical Turk. They find that according to the human study, their model is able to both answer the question correctly and generate a valid explanation about 60.5% of the time.

**Strengths:**

The main strength of this paper is the model's empirical performance: it outperforms the baseline and answers the question correctly while generating a valid explanation about 60.5% of the time.

**Weaknesses:**

One weakness of the paper is its lack of technical novelty, as the main algorithm is a LSTM/Transformer Decoder architecture which sits on top of the state-of-the-art VQA model. A potentially bigger concern is the empirical results: specifically the lack of baselines. In particular, the authors only compare against one baseline, out of several related algorithms in Section 2. Additionally, no baseline explanations are evaluated by humans in the case study.

**Overall Recommendation:**

While the paper fits the theme of the workshop, I would recommend rejection due to the lack of technical novelty and an inadequate comparison with baseline models. However, I must admit that I am not very familiar with this line of work, so I may have missed something in my evaluation of the authors' model.

**Review Confidence:**

2: The reviewer is willing to defend the evaluation, but it is quite likely that the reviewer did not understand central parts of the paper

---

### Official Review · Reviewer_xAzC · 2022-10-20
**An interesting VQA framework providing good performance and interpretability**

**Overall Rating:** 7

**Summary:**

The paper proposes to learn an explanation generator which is added to a VQA framework with human annotation as supervision. Experiments show that the proposed new framework achieves comparable results compared to the baseline framework without the explaining ability. The human study further shows that the generated explanations are valid and can lead to answers.



**Strengths:**

1. The studied problem is important as an interpretable VQA system is more reliable and easier to inspect.
2. The proposed method maintains the performance by setting the explanation generator as an added component which is trained together with downstream tasks as multi-task learning.
3. The paper is well-organized and easy to follow.

**Weaknesses:**

1. The proposed framework may treat task prediction and explanation generation as two independent tasks and the resulting explanations may not faithfully represent the underlying reasoning process. Perhaps conditioning the task prediction on the explanations will be a more reasonable design. The authors should also conduct further analysis to validate the faithfulness of the explanations.
2. The fact that adding explanations as regularization does not offer significant performance gain makes it questionable to build such a multi-task system. Perhaps some robustness analyses will be helpful to demonstrate the advantages of the proposed method.
3. Since the GQA-REX dataset provides annotations on what objects are important in deciding the answer, the authors can indeed leverage such annotations to investigate whether the model is attending to the right features instead of using the word-overlap based metrics (BLEU, ROUGE, etc.).

**Overall Recommendation:**

I recommend accepting the paper as it presents a VQA framework which offers human-readable explanations while maintaining the performance. But I encouraged the authors to further validate the faithfulness of the explanations since they are not part of the decision-making process.

**Review Confidence:**

4: The reviewer is confident but not absolutely certain that the evaluation is correct

---

### Official Review · Reviewer_1Xje · 2022-10-21
**Interesting topic, limited novelty**

**Overall Rating:** 6

**Summary:**

This work studies explanation generation in visual question answering (VQA). Their model is built upon the coarse-to-fine reasoning (CFR) architecture, and they added an LSTM or Transformer decoder to generate explanations. They evaluate their models on VQA-E and GQA-REX datasets, and show that their approach generates better explanations compared to the baseline, while still keeps a high VQA accuracy. Meanwhile, they conducted a human study, and show that most of the generated explanations are valid.

**Strengths:**

Generate explanations for visual question answering is an interesting topic. Also, the human study shows some correlations between explanation validity and prediction correctness, though the conclusions are not surprising.

**Weaknesses:**

The technical novelty is limited: the datasets are from prior work, and they simply add a decoder to one prior model architecture for VQA. Overall, the message of this paper is unclear, as there are already prior works showing that we can train a model to generate explanations for VQA, and some existing works show that training with explanations improve the prediction accuracy for the VQA task. It is more interesting to propose a better way to leverage explanations and further improve over the SOTA results.

**Overall Recommendation:**

The paper studies an interesting topic that is relevant to the workshop, but the technical novelty is limited. Borderline accept.

**Review Confidence:**

5: The reviewer is absolutely certain that the evaluation is correct and very familiar with the relevant literature

---

### Decision · Program_Chairs · 2022-10-23

**Decision:**

Accept

**Comment:**

This paper aims to make the VQA task more explainable and interpretable. The contents are relevant to this workshop and the empirical results are interesting.